# Cervical cytology intelligent diagnosis based on object detection technology

**Meiquan Xu**
Shenzhen Second People's Hospital.
xumeiquan@126.com

**Weixiu Zeng**
Semptian Co., Ltd. Machine Learning Lab.
zengweixiu@gmail.com

**Yanhua Sun**
Shenzhen Second People's Hospital.
syhbeibei@126.com

**Hunhui Wu**
Shenzhen Second People's Hospital.
736886978@qq.com

**Tingting Wu**
Shenzhen Second People's Hospital.
tingting8328@163.com

**Yajie Yang**
Shenzhen Second People's Hospital.
sz_yajieyang@sina.com

**Meng Zhang**
Shenzhen Second People's Hospital.
mmzmzgd@163.com

**Zeji Zhu**
Semptian Co., Ltd. Machine Learning Lab.
zhuzeji@126.com

**Longsen Chen**
Semptian Co., Ltd. Machine Learning Lab.
chenlongsen@semptian.cn

## Abstract

Objectives: (i)Explore a new method for applying artificial intelligence to cervical cytology diagnosis. (ii)Realize an automatic detection system of positive cervical squamous epithelial cell. Methods: (i) The method can be divided into two phases: training and testing. (ii) For the training phases, first of all, we collect 500 cervical cell slides. Through scanning, annotating, and extracting patches, we get the training set, validation set and test set. Then, based on the Faster R-CNN (Faster Regions with CNN features) [4], we construct a neural network model for cell detection and classification. After continuous training, validation, analysis and tuning, we finally get a well trained neural network model. (iii) During the testing phase, we use the previously well trained neural model to predict the test set which consists of 100 whole slide images of cervical cytology. The model detects the five types of target cells at first, and then counts the number of cells in each category, finally, generates a diagnosis. Results: The positive precision rate on the validation set is 0.91. On the test set, for two-class problem,the accuracy is 0.78. For four-class problem, the accuracy is 0.70. Conclusion: Object detection technology has unique advantages in applying to cervical cytology. Through accurate detection and classification of various types of abnormal cells, as well as the statistics of each category, a comprehensive conclusion is made. This idea adopted in this study accords with doctors' traditional diagnosis process to a certain degree. Results show that the intelligent system realized with deep learning technology has the advantages of high speed, high consistency, and well diagnostic performance.

1st Conference on Medical Imaging with Deep Learning (MIDL 2018), Amsterdam, The Netherlands.

# 1 Introduction

Cervical cancer is the most common gynecologic malignant tumor, but cervical cancer is not an incurable disease. If it can be found early, the cure rate of the disease is very high. Therefore, early examination and early discovery are very important. The application of thin liquid-based cytology test has greatly improved the positive diagnosis rate of cervical lesions, and has became one of the most important measures to reduce the incidence of cervical cancer [1]. Currently, cervical cancer screening in China adopts a three-steps process: the first step is cytology, the second step is colposcopy, the final step is histologic biopsy [1].

Cytological examination has an important position in the three-steps process of cervical cancer screening, but it also faces the following serious problems: (i) The cytological examination presents a situation of large demand but lack of pathological resources. The lack of professional cell pathology expert maybe become a bottleneck of cervical cancer screening. (ii) Due to the difference in the level of pathologists, the positive coincidence rate of manual reading is not high and fluctuating. (iii)Traditional way of reading slide under the microscope in clinical diagnosis is not convenient to review and therefore results in reducing the sensitivity and specificity of screening.

The application of artificial intelligence in cervical cytology is expected to solve these problems, due to its features of high speed, high repeatability, and strong visual effects.

This study follows the 2014 TBS reporting system. The squamous intraepithelial lesion is divided into low squamous intraepithelial lesion (LSIL) and high squamous intraepithelial lesion (HSIL). LSIL is diagnosed by detecting typical low-grade lesion cells and counting the number of it. HSIL is diagnosed by detecting typical highly lesion cells or cell clusters and counting their number. For atypical squamous epithelial cells, the TBS reporting system divides it into atypical squamous cells of undetermined significance (ASC-US) and atypical squamous cell-cannot exclude HSIL (ASC-H) [1]. However, due to the lack of ASC-H cell samples, HSIL cell is not target cell. We only detect ASC-US cells, and count the number of it to diagnose the ASC-US. For the sample satisfaction evaluation section, it is prompted by detecting endocervical cell (EC) and metaplastic squamous cell (MSC).

The flow diagram of the entire research is shown in figure 1 below:

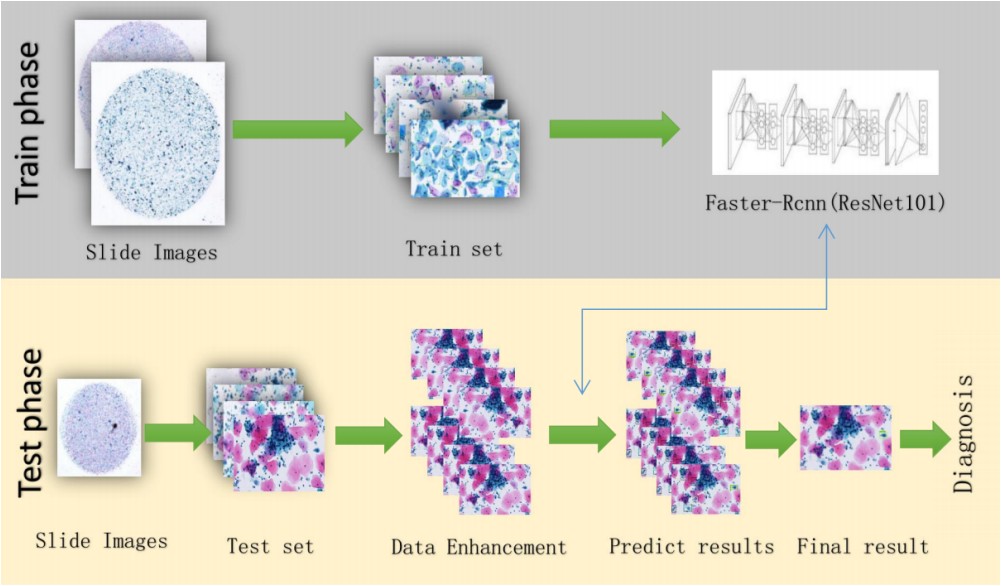

Figure 1: System Flow Diagram

# 2  Method

## 2.1  Making Data Set

500 samples of cervical cytology stained with Pap are collected from the pathology department of shenzhen Second People's Hospital, including 450 positive cases and 50 negative cases. All slides are scanned at 20X, using a digital slide scanner of Kfbio to obtain 500 whole slide images. 50 positive samples and 50 negative samples are selected from these 500 samples to serve as the test set. The remaining 400 positive samples serve as the source of training set and validation set.

### 2.1.1  Making Training Set and Validation Set

Step 1: Organize experts to annotate target cells as many as possible on the whole slide images of 400 positive samples, including ASC-US, LSIL, HSIL, EC and MSC, a total five kinds of cells .

Step 2: On the whole slide image, we cut rectangular areas of 1024*600 in grid. If an area contains target cells, the area would be saved as a picture, an annotation file based on the picture coordinates is generated at the same time which records the specific coordinates of each cell annotated by an expert. According to this method, we get a total of 5,721 pictures and their corresponding annotation files.

Step 3: we select 500 pictures randomly from 5721 pictures as the validation set and the rest 5221 pictures are used to the training set.

Step 4: Organize experts to review 5721 pictures including the training set and the validation set to ensure that the annotated cells are accurately classified, and in these pictures, there is no ASC-US, LSIL, HSIL, EC or MSC cells are ignored.

Step 5: Make a statistics for training set and validation set. The statistics of five kinds of cells included in the training set are shown in Table 1 below:

Table 1: Statistics of the various types of cells in the training set

| Cell Type | Amount | Ratio (%) |
|-----------|--------|-----------|
| ASC-US    | 1962   | 21.2      |
| LSIL      | 860    | 9.3       |
| HSIL      | 939    | 10.2      |
| EC        | 3589   | 38.8      |
| MSC       | 1896   | 20.5      |
| Total     | 9246   | 100       |

The statistics of five kinds of cells included in the validation set are shown in Table 2 below:

Table 2: Statistics of the various types of cells in the validation set

| Cell Type | Amount | Ratio (%) |
|-----------|--------|-----------|
| ASC-US    | 235    | 22.1      |
| LSIL      | 90     | 8.5       |
| HSIL      | 141    | 13.3      |
| EC        | 311    | 29.3      |
| MSC       | 284    | 26.8      |
| Total     | 1061   | 100       |

### 2.1.2  Making Test Set

Step 1: Test set is made up of 100 whole slide images. We firstly separate the foreground area (cell area) and background area of each image in the test set.

Step 2: Through cutting the foreground area into certain number of pictures with a size of 1024*600 in grid, each image can generate different numbers of 1024*600 pictures depending on the size of the foreground area.

### 2.1.3 Foreground Extraction

According to the characteristics of cervical cytology slide's production process, cells in the slide are generally distributed within a circle. In the scanning process, the scanning area is generally selected manually, and a larger rectangular area than the circumscribed rectangle of the circle is selected for scanning. Therefore, the scanned image contains a part of none cell area. These none cell areas are called background areas. Removing the background area can reduce the computational cost and short the time required for intelligent diagnosis.

In the cervical cytology whole slide image, the cells in the foreground region are stained with Pap, which is rich in color and the background area is nearly white. After trying different color space comparisons, the Z channel of the XYZ color space is suggested for foreground extraction in this study. If the Z channel value of a pixel is less than 0.9, it is considered as a foreground pixel. Similarly, if the Z channel value of a pixel is larger than 0.9, this pixel is considered as a background pixel. XYZ color space is a kind of device independent standard color space. It is often used as a benchmark color description, that is to say, any color can be synthesized using XYZ three primary colors. Experiments show that using Z channel can effectively distinguish foreground and background.

The effect is shown in Figure 2 below:

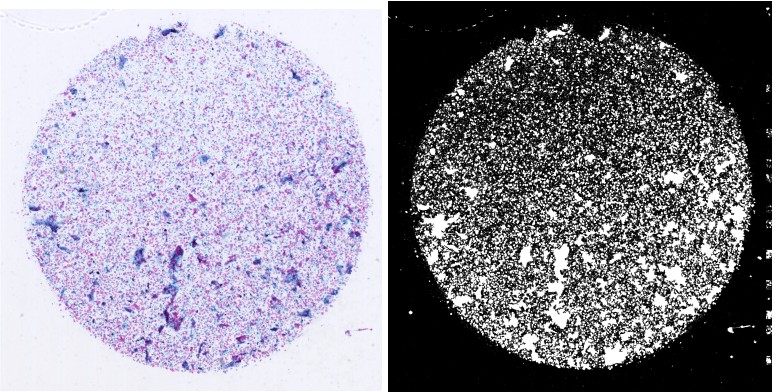

Figure 2: Foreground Extraction

Through calculations, after the extraction of the foreground, the computation amount of each whole slide image can be reduced by about 10%.

## 2.2 Model Training

The basic network structure used in this study is the Faster R-CNN object detection network. The Faster R-CNN network was developed from R-CNN and Fast R-CNN. Its basic principle of the network is the region-proposal algorithm. Through constant adjustment and optimization of network structure, the feature extraction, region proposal, region classification and position correction are unified into one deep network. As a result, this structure can avoid the repeated calculation and improve the detection efficiency [4] [5].

The basic work flow of the network is inputting pictures, extracting features of input pictures, generating feature map, according to the feature map to propose regions, classifying regions, and finally correcting the position [6].

The feature extraction network used in this study is resnet101.we can also use other classification networks. The feature map is got from the input picture after several convolution layers. RPN network use 3 scales(128*128, 256*256, 512*512) and 3 aspect ratios(1:1, 1:2, 2:1) to generate 9 anchors at each position of the feature map. Trained RPN network can divide these proposed regions into target regions and false target regions, the false target regions are abandoned and the target regions are sent to the subsequent classifier network to get the final classification.

## 2.3 Evaluation and Results

This study use the precision(P) and recall(R) to assess the trained model's ability at the cell level.

True Positive (TP) means an object detected by model is consistent with expert. In this study, to considered to be consistent should meeting two requirements. The one is the position and border of the box are consistent, and the second requirement is the category of the box should be same too. In the actual evaluation, we access the consistency of a detected rectangle box and a ground truth rectangle box by IOU, which is the intersection of these two rectangular boxes divided by their union [7]. The formula is defined as follows:

$$IOU = \frac{detection \cap ground\_truth}{detection \cup ground\_truth}$$

When the IOU of a detected box and a ground truth box is larger than 0.5, and the category classified by the model is the same as the ground truth, this detected box is considered to be TP; otherwise, the detected box is considered to be false positive (FP).

Taking the ASC-US as an example, the precision rate indicates the accuracy of the ASC-US cells detected by the model, and the ASC-US precision rate equals the number of detected ASC-US cells that matches the expert's annotation (TP) divided by the total number of ASC-US cells detected (TP + FP). The recall shows the degree of leak detection. The recall rate of ASC-US equals the number of detected ASC-US cells that matches the doctor's annotation (TP) devided by the total number of ASC-US cells annotated by the expert(TP+FN). The calculation formulas of other classes are defined similarly to this.

According to the above evaluation method, the results of the study on the validation set are shown in Table 3 below:

Table 3: Evaluation results of validation Set

| Cell Type | TP | TP+FP | TP+FN | P | R |
|-----------|-----|-------|-------|--------|--------|
| ASC-US | 123 | 166 | 235 | 0.7410 | 0.5234 |
| LSIL | 45 | 54 | 90 | 0.8333 | 0.5000 |
| HSIL | 63 | 72 | 141 | 0.8750 | 0.4468 |
| EC | 151 | 161 | 311 | 0.9379 | 0.4855 |
| MSC | 76 | 77 | 28 | 0.9870 | 0.2676 |
| Positive | 268 | 292 | 466 | 0.9178 | 0.5751 |

This study use accuracy to evaluate the diagnostic capabilities of the model at the whole slide image level. Accuracy equals the number of slides that the model's diagnosis is consistent with the expert's diagnosis divided by the total number of test slides .

Finally, on the test set composed of 100 whole slide images, the accuracy of the four classifications is 0.70, and the accuracy of the two classifications is 0.78.

## 2.4 Tuning

This study adopts data enhancement during the test process. A picture is rotated and mirrored to form a group of eight pictures.Therefore, a picture is actually predicted eight times. And then, a corresponding result fusion technique is used to combine eight predict results into one final result. Results show that data enhancement can improve the diagnostic capability of the model. The specific method is described as follows:

### 2.4.1 Data Enhancement

Each patch is rotated 90 degrees, 180 degrees, 270 degrees, mirrored, mirrored and rotated 90 degrees, 180 degrees, 270 degrees, to form a group of eight patches, including the original patch. The four of the group of patches enhanced from a patch are shown in Figure 3 :

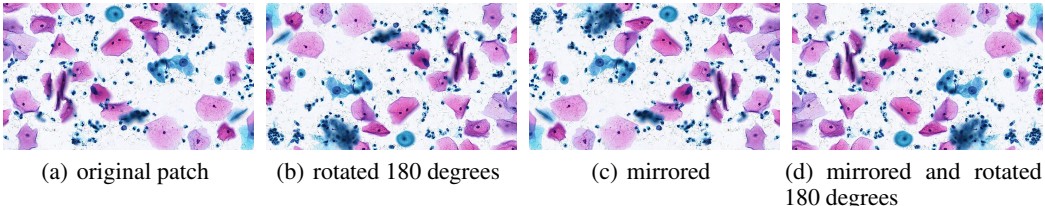

| (a) original patch | (b) rotated 180 degrees | (c) mirrored | (d) mirrored and rotated 180 degrees |

Figure 3: Tne four of a group of patches enhanced by one patch

### 2.4.2 Result Fusion

A group of eight patches enhanced from one patch have eight predict results after network prediction. The experimental results show that there is a certain difference between them. The figure 4 below shows the four of the eight results.

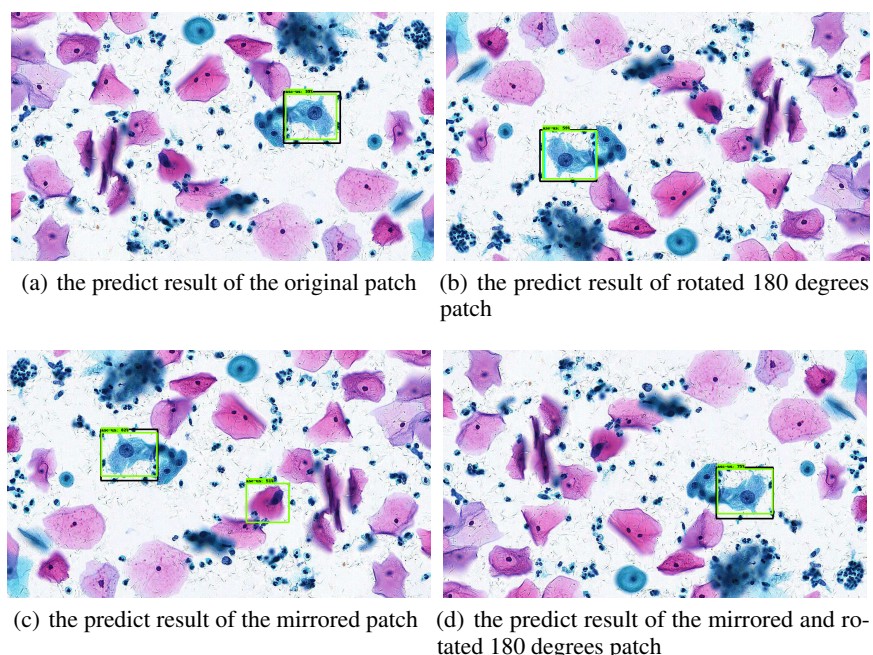

(a) the predict result of the original patch  (b) the predict result of rotated 180 degrees patch

(c) the predict result of the mirrored patch  (d) the predict result of the mirrored and rotated 180 degrees patch

Figure 4: The four of the eight results.

The result fusion technology firstly calculates the IOU of any two detected boxes in the eight predict results of one original picture. If IOU is larger than 0.6, the two detected boxes are considered to be a same object. Finally, for an original picture, we can get all the objects which are detected through eight prediction. What's more, we get the number of times(N) each object was detected during the eight prediction.

For the score of each detected object, a penalty term (a+b*N) linearly associated with N is introduced. The final score S of each object defines the sum of the scores of N times divided by the penalty term. The specific formula is defined as follows:

$$S = \frac{\sum_1^n score}{a + b * n}$$

When a=0 and b=1, this formula is equal to calculating the average score of N times of one object directly. Adjusting the slope b>1,and a>0, can realize that when n is small, the final score is lower than the average score. A threshold value is used to filter out the detected objects with a small score. That is, on the one hand, the object with a low score is removed, although it maybe recognized

multiple times, on the other hand, the object with a low detected frequency is removed, although it maybe has high test score on occasions.

### 2.4.3 Contrast

On the validation set consisted of 500 patches, the result with data enhancement and the result without data enhancement are shown in Table 4 and Table 5 below:

Table 4: Comparison between the result with data enhancement and the result without data enhancement,(N) indicates no data enhancement

| Cell Type | P | R | P(N) | R(N) |
|---|---|---|---|---|
| ASC-US | 0.7410 | 0.5234 | 0.6503 | 0.5064 |
| LSIL | 0.8333 | 0.5000 | 0.7385 | 0.5330 |
| HSIL | 0.8750 | 0.4468 | 0.6988 | 0.4113 |
| EC | 0.9397 | 0.4855 | 0.7928 | 0.5659 |
| MSC | 0.9870 | 0.2676 | 0.7279 | 0.3768 |
| Positive | 0.9178 | 0.5751 | 0.7964 | 0.5644 |

Table 5: Comparison between the result with data enhancement and the result without data enhancement,(N) indicates no data enhancement

| Cell Type | TP | FP+TP | TP(N) | TP+FP(N) |
|---|---|---|---|---|
| ASC-US | 123 | 166 | 119 | 183 |
| LSIL | 45 | 54 | 48 | 65 |
| HSIL | 63 | 72 | 58 | 83 |
| EC | 151 | 161 | 176 | 222 |
| MSC | 76 | 77 | 107 | 147 |
| Positive | 268 | 292 | 263 | 331 |

The data in the table shows that the data enhancement and result fusion technology can reducing the number of FP, while remain the number of TP. As a result, the overall recall and precision are increased.

## 3 Conclusion

This study explored a new method for applying artificial intelligence to cervical cytology. An automatic detection system of positive cervical squamous epithelial cell was realized based on object detection technology. The system can detect three kinds of abnormal cell in the whole slide image, count the specific number of each category, and proposal a final diagnosis of the slide. The results of the study indicate that this method can automatic diagnosis squamous intraepithelial lesion, and report sample satisfaction by giving the number of EC and MSC of a whole slide image.

The whole system has good detection precision, high efficiency, and well intuitionistic.

## 4 Discussion

Based on cell classification, the study on the task of using deep learning technology to achieve cervical cytology detection and identification has been explored all around the world[5][6]. Bilal Taha et al. has analyzed the Pap smear image to detect cervical cancer based on deep learning method in 2017. The pre-trained CNN architecture was used as the feature extractor and the output feature was used as the input of the training SVM classifier. The result indicates that deep learning can effectively screen cervical cancer. However, this study performs well only on the task of classifying individual cell pictures, and their use on whole slide images is limited by cell or cell cluster's detection and location. What's more, cell segmentation technology based on tradition image processing algorithm is not practice in clinic, because of it is time consuming and lack of generalization.

In the early stage of our research, we also established a cell classification system with Google Inception V3[9] model. The input patch size of the classification system was fixed, we adopted a grid sliding window on whole slide images to cut patches as input, then the characteristics of the input patch were extracted. Model classify a patch through these features. But the sliding window with a fixed size can not accurately included a cell or a cell cluster. Most windows include an incomplete cell and some windows include none cell. For basal cell which is relatively small, the fixed patch size is too large, for surface squamous epithelial cells or cell cluster, the fixed patch size seems small.

In the whole slide image, cell amount is very large, overlapping is also difficult and common, in addition, the target cells have various size, but the general classification model's input size is fixed. These factors result in a simple classification model is not suitable to cytology.

The biggest innovation of this research is applying the object detection model Faster R-CNN to cytology. This method can solve the difficulties of overlapping and various size of target cells when applying artificial intelligence to cervical cytology.

The object detection technology, on the one hand, proposes regions with different scales and different ratios, so that either surface squamous epithelial cells and cell cluster with large size, or basal cells with small size can be detected, on the other hand, it Integrates detection and classification.

Data enhancement and result fusion techniques used in this study, through predicting 8 copies of a patch, and the introduction of a penalty term, can greatly inhibit false positives. Compared to the case of none data enhancement, the precision of positive cell increases by about 15%, when the recall of positive cell almost remains the same.

The foreground extraction technology also reduces the amount of computation by about 10% substantively, and controls the entire diagnostic time which including detection, classification,statistics and final diagnoses of the whole slide image within 5 minutes.

In the experimental results, the precision of positive cell is 0.91, and the precision of ASC-US, LSIL, and HSIL are relatively low. ASC-US, LSIL, and HSIL are collectively referred to as positive cell. This result indicates that the model can effectively differentiate between positive and negative cell, but it has certain difficulties in distinguishing ASC-US, LSIL, and HSIL. This also reflects the morphologically confusing features of ASC-US, LSIL, and HSIL at the level of single cells or cell clusters.

In the experimental results, the accuracy of diagnosing a whole slide image into positive or negative is 0.78. That means 78 cases are diagnosed correctly, and 22 cases are given wrong diagnosis results. Among these wrong samples, 10 are false positives, 12 are false negatives. In the false positive cases, 1 case is moderate inflammation, 2 cases are epidermal cell atrophy, and 3 cases are fungal infection. This result shows that morphological abnormality caused by inflammation, epidermal cell atrophy, and fungal infections is one of the major types of system diagnostic error.

In cytology, the morphology of positive cells is varied, and it is still a challenging problem for AI to be not ignore any abnormal cell. However, in clinical applications, if a HSIL case is missed, it means that a patient who is likely to develop cervical cancer would lose the opportunity for treatment. Therefore, artificial intelligence should pay more attention to negative precision rate, and leaves the positive and suspicious positive to a more professional doctor for further review when applying to cytology.

Therefore, the follow-up research direction will include the following points: (1) Pay more attention to the labeling standards of various training samples. Accurate and uniform labeled training set is prerequisite for training an effective model. For example, if the features of ASC-US, LSIL, and HSIL are cleared strictly on the training set, the trained model would have a better performance in distinguish these types of cells. On the contrary, if the training set is fuzzy, the ability of the model would be reduced. (2) Extend the class of detection target cell, such as microorganisms, abnormal gland cells, etc., to more fully collect the effective information on the whole slide image, which would lead to a more accurate final diagnosis. (3) The number of training set used in this study is relatively small. Compared with EC and MSC, which are very common in the whole slide image, the collection of abnormal cells is more difficult, so that the proportion of these abnormal cells in the training set is small. Subsequent research will expand the number of training sets and balance the proportion of each category. (4) Focus on negative precision rate and negative recall rate. The negative precision rate is a measure indicator with more clinically practical value.

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
