# OpenReview forum: "Cervical Cytology Intelligent Diagnosis Based On Object Detection Technology"
_MIDL.amsterdam/2018/Conference — Submitted to MIDL 2018_

### Review · AnonReviewer3 · 2018-05-04
**relevant problem, relevant architecture, limited experimental evaluation**

**Rating:** 2
**Confidence:** 2

**Review:**

The paper proposes to use Faster RCNN for cervical cytology diagnosis.

pros
+ relevant problem, relevant architecture

cons
- weak motivation for the model of choice, limited experimental evaluation, clarity could be improved

*The paper presentation could be significantly improved.
*The choice of Faster RCNN should be properly motivated, why did you choose this model instead of other more recent approaches such as YOLO or MaskRCNN?
*Data preparation: Section 2 states that 500 samples are gathered, 450 positive cases and 50 negative cases. Then,50 positive and 50 negative samples are selected from these 500 as test set. Does this mean that the training set only has positive samples?
*What the influence on the 0.9 threshold to extract foreground and background?
*It would be appropriate to include a more detailed review of Faster RCNN.
*RPN is not introduced in the text.
*Why only report accuracy on the test set and not other metrics reported in validation? (page 5)
*Table 3: having 3 columns for TP, FP and FN would be beneficial (instead of TP, TP+FP, TP+FN).
*The method is not compared to any baseline, making it hard to assess the contribution.
*Is data augmentation applied during training?
*The so-called data enhancement strategy has been widely used in the literature, please reference some works.
*Why is the overlapping threshold of the data enhancement part set to 0.6?
*Do results on table 3 correspond to data enhancement?
*Tables 4 and 5 could be merged.
*The authors reference literature on the same topic using deep learning, it would be important to provide results of these other methods as baselines.
*If experiments were performed using Inception V3 as well, it would be good to report results.
*The discussion claims that ASC-US, LSIL and HSIL are referred to as positive cells. Why do the authors think the taking into account independent classes leads to low performance? How could this be improved?

**Special Issue:**

No

---

> ### Comment · ~zeng_weixiu2 · 2018-05-14
> **interpretation of some question**
>
> Thank you so much for the valuable comment.
> The responds to the reviewer’s comments are as follows:
> *We really should try more models such as YOLO and MaskRCNN. But we think the model is not the main factor affecting the final accuracy in this problem. So, we don't  pay enough attention to the model contrast.
> *The training set is exactly only extracted from positive samples. In fact, a positive sample means a whole slide image of a positive case, which includes many positive cells and negative cells at the same time.
> *we really should add more detailed review of our model.
> * The validation aims at cells, but test set aims at cases. therefore, the metric is designed different.
> *Previous works of applying deep learning to cervical cytology are not based on whole slide image, but single cell pictures. Therefore, the evaluate result is not comparable.
> *Data augmentation  is not applied during training.
> *The result on Table 3 correspond to data enhancement, and Table 4 and 5 is a contrast of data enhancement and no  data enhancement .
> * Taking into account independent classes leads to low performance means the precision of each classes would be lower, but if we refer all classes as positive, the errors between classes will be ignored. When we prepare training set, distinguish single cell into independent classes is sometimes difficult. To solve this problem, we think we need mode data and more professional annotation standards.
> *Some comments of reviewer is very helpful for us, and  we will continue to improve.

---

### Review · AnonReviewer2 · 2018-05-09
**This paper presents cervical cytology diagnosis using object detection framework, Faster R-CNN. The authors have applied the Faster R-CNN to new cervical data and did not make any performance comparison to the other baseline methods.**

**Rating:** 2
**Confidence:** 3

**Review:**


Quality & Clarity

#1. The organization of this paper is not good and need to be re-organized.
#2. The result section of this paper should be clarified.

Originality & Significance

(+) The problem definition was interesting.
(-) There is no technical novelty.
(-) There is no baseline method. Authors should compare to other methods.
(-) The details for model training is not included in this paper.
(-) Tuning section is only general data augmentation scheme.

**Special Issue:**

No

---

> ### Comment · ~zeng_weixiu2 · 2018-05-14
> **Thanks for precious comment**
>
> Thank you very much for your pertinent comments.
> Because the baseline method of applying deep learning in cervical cytology pay more attention on cell classification,  but not a combination of detection and classification,  therefore, the evaluate result is not comparable.

---

### Review · AnonReviewer1 · 2018-05-09
**new application of faster R-CNN**

**Rating:** 1
**Confidence:** 3

**Review:**

The work explores the use of faster R-CNN for automatic detection in cervical cytology.

Although this is a good study applying a recent deep learning method to a new application, I am not sure this is a good fit to the MIDL conference. The pipeline including pre-processing is very specific to the particular data, and since the methodology itself is not novel, it is a bit unclear what to learn from the paper.

I believe it would be a better fit to a more application focused venue or journal.




**Special Issue:**

No

---

> ### Comment · ~zeng_weixiu2 · 2018-05-14
> **Explore applying faster R-CNN to cervical cytology diagnosis**
>
> Hi,
> Thank the reviewer very much for these precious comments and suggestions.
> In my opinion, this paper is not just a new application of faster R-CNN, but a method introduction of applying faster R-CNN to cervical cytology diagnosis. It focus on define the problem of  cervical cytology diagnosis. We think  the problem is a combine of detection and classification, which is different to the previous definition of only cell classification.
> So, this paper fits to the topic of medical image analysis using deep learning.

---

### Decision · Program_Chairs · 2018-05-15
**Paper19 Acceptance Decision**

Reject